# Rise of Constabulary Maritime Agencies in Southeast Asia: Vietnam's Paragunboat Diplomacy in the North Natuna Seas

Bama Andika Putra [1,2] 

1    School of Sociology, Politics, and International Studies, University of Bristol, Bristol BS8 1QU, UK;
     bama.putra@bristol.ac.uk
2    Department of International Relations, Universitas Hasanuddin, Makassar 90245, Indonesia;
     bama@unhas.ac.id

**Abstract:** The rising tensions in disputed waters in Southeast Asia have caused policymakers to diverge their maritime diplomatic strategy to include maritime constabulary forces. The use of coastguards and other non-military vessels are an emerging trend in the maritime diplomatic strategy of Southeast Asian states, including in the high-profile case of the North Natuna Seas, to which scholars pay little attention. This article contends that (1) contemporary maritime diplomacy in Southeast Asia positions the utilization of maritime constabulary forces (coastguards, maritime law enforcement agencies) as its primary maritime diplomatic strategy; (2) Vietnam's coercive turn in its maritime disputed areas was a deliberate attempt to balance a coercive-cooperative stance against Indonesia in the North Natuna Seas, following its traditional coercive maritime diplomatic stance against China, and; (3) Vietnam's utilization of maritime constabulary forces as a measure to solidify its sovereign claims coincided with the benefits of tactical military flexibility and non-escalatory means to achieve its aims in the Natuna Seas. This empirical explanatory research delves into the development of Vietnam's coastguards and maritime law enforcement agencies by interpreting the secondary data from the Indonesia Ocean Justice Initiative between 2021–2022 on cases relating to suspicious maneuvers conducted by the Vietnamese Fisheries Resource Surveillance vessels safeguarding the conduct of Vietnamese IUUF.

**Keywords:** paragunboat diplomacy; maritime constabulary forces; maritime law enforcement agencies; maritime diplomacy; Vietnam; North Natuna Sea



## 1. Introduction

Neighboring states continue to challenge Indonesia's claim over the North Natuna Seas. With China's growing assertiveness in the South China Seas and the use of alternative maritime forces to solidify its nine-dash line claims, policymakers in Indonesia have started to mobilize additional resources and attention to counter aggressions in the disputed waters (Putra 2022; Putra and Cangara 2022; Pattiradjawane and Soebagjo 2015; Meyer et al. 2019). Nevertheless, China is not the only country where Indonesia faces tensions within the North Natuna Sea. The overlapping exclusive economic zone (EEZ) claim between Indonesia and Vietnam has aggravated the bilateral relations between the two Association of Southeast Asian Nations (ASEAN) members for two decades (Putra 2020).

Based on the United Nations Convention on the Law of the Sea (UNCLOS), parts of the North Natuna Sea can be claimed by Indonesia, Vietnam, and China due to the overlapping EEZ borders. As a result, the three countries have intensified their power projections in those waters by employing divergent strategies to show effective occupancy (Tiola 2020; Suryadinata 2016; JP 2020). China, for example, is well known for its highly militarized fishing fleets and coastguards operating suspicious maneuvers (Giang 2018; de Castro 2022). Indonesia's priority over the contested waters is based on Jokowi's intentions to end the 'who is Indonesia's coast guard' question by announcing that the Indonesian Maritime Security Agency, created in 2014, will act as Indonesia's official coastguard (BAKAMLA 2021).

BAKAMLA's employment in the North Natuna Seas has immensely impacted Indonesia's decisive posture and resolve. On the other hand, Vietnam is showing a different pattern in its actions. Vietnam prioritizes using maritime constabulary forces as the primary agent to safeguard Vietnam's interests and solidify sovereign claims in the disputed waters.

The overlapping EEZ has strained Vietnam and Indonesia's bilateral relation claims that they share. Due to the inconclusive borders, IUUF has become common in the Natuna waters. Unfortunately, most perpetrators are Vietnamese fishing boats. In response to this, Indonesia has increased its presence in the sea by employing multiple stakeholders to safeguard Indonesia's waters. Specifically, Indonesia has employed the following institutions to protect Indonesia's waters: BAKAMLA, Indonesia's Ministry of Marine Affairs and Fisheries (KKP), Indonesian Police (POLRI), Indonesian Customs, Indonesian Ministry of Transportation (SCG), and the Indonesian Navy.

Vietnam's maritime constabulary forces in the North Natuna Seas coincide with its decisive power projection in other parts of the South China Sea. In the Spratly Islands, Vietnam's tensions with China have led it to undertake a similar land reclamation strategy in those small islands (AMTI 2021). Additionally, it has persisted in demonstrating to China that Vietnam's EEZ cannot be compromised and that it is unwilling to renounce its sovereign claims to China.

Vietnam's power projection in the South China Sea aligns with Vietnam's hedging policies vis-à-vis great powers in the Indo-Pacific. The seas are turbulent in both Southeast Asia and in the larger Indo-Pacific. Maritime boundary tensions highlight the region's geopolitics, undermining possible cooperation that could emerge due to the lucrative resources that remain unexplored in the Indo-Pacific seas (Tan 2020; Chacko and Willis 2018). Vietnam's management of international relations via sea thus concludes that a non-military response to existing maritime border disputes is ideal as a form of Vietnam's protection. Like most Southeast Asian nations, and similar to the Indo-Pacific great power rivalry between the US and China, Vietnam does not want to directly align itself with a great power to minimize the risk of choosing the wrong side (Kuik 2022). Vietnam's maritime diplomatic policy is developed in light of this conundrum, as is the need to adopt a course of action that would demonstrate a firm but restrained use of coercion while avoiding escalatory levels of force and, as a result, justifying Vietnam's paragunboat diplomacy in the North Natuna Seas.

This article argues that Vietnam's contemporary maritime diplomatic strategy in the overlapping EEZ area of the North Natuna Seas is through its maritime constabulary agencies. These units can be categorized as non-military, civilian fleets designed to carry out various tasks at sea. Some vessels under specific ministries are tasked with countering non-traditional security threats, while some focus on stopping illegal, unregulated, and unreported fishing (IUUF). Nevertheless, Vietnam's strategy has been to use the Vietnamese Fisheries Resource Surveillance (VFRS) operating under the mandate of Vietnam's Ministry of Agriculture and Rural Development. VFRSs active presence in the North Natuna Seas is Vietnam's attempt to solidify its sovereign claims in the disputed waters by escorting local Vietnamese fishing boats, conducting suspicious maneuvers in the borders of Indonesia's EEZ, and actively engaging in a limited-coercive manner with Indonesian maritime officials. A representation of the activities undertaken by VFRS is framed as 'paragunboat diplomacy,' a term coined by Le Miere in 2014.

This article thus focuses on the empirical puzzle of Vietnam's contemporary maritime diplomatic strategy against Indonesia. Specifically, it questions why Vietnam utilizes maritime constabulary forces in facing maritime disputes in the North Natuna Seas against Indonesia. The problem is grounded by the fact that the active presence of the Vietnamese navy in the troubled waters has been stagnant in recent years. However, this article highlights the coercive maneuvers of Vietnamese maritime constabulary forces over recent years.

In doing so, this empirical explanatory research utilizes secondary data from the Indonesian Ocean Justice Initiative (IOJI) between 2021–2022. The timeframe of 2021–2022

is the focus of this research as the data are only available within these two years, coinciding with Vietnam's growing coercive maneuvers in the North Natuna Seas. Meanwhile, empirical explanatory research is utilized as it allows for greater development of arguments based on primary data attained from the IOJI. Based on the data from the satellite imagery of Sentinel-2 and an automatic identification system, we can translate the meanings of certain maneuvers made by the VFRS, leading to a determination of its actual intentions and purposes. By revealing patterns of Vietnam's maritime diplomatic strategies, we can argue the effective use of Vietnam's paragunboat diplomacy in the North Natuna Seas.

## 2. Constabulary Maritime Forces in Maritime Diplomacy

Maritime constabulary forces/agencies represent a variety of state-owned actors at sea. The actors could be coastguards, maritime paramilitary agencies, fisheries surveillance, fishing militia, and other maritime law-enforcing stakeholders. A common characteristic that unites the different actors is the non-military nature of the fleets operated, acting more as a civilian ship with certain defined tasks. The utilization of these fleets, by conducting limited-coercive maneuvers to represent the interests of the state, is termed 'paragunboat diplomacy' (Le Mière 2014). Paragunboat diplomacy combines two distinct words, 'paramilitary' and 'gunboat,' to describe a maritime diplomatic strategy to manage international relations. Though paramilitary is well identified, gunboat diplomacy requires further elaboration.

Gunboat diplomacy, or 'naval diplomacy,' simply uses naval forces to manage international relations via sea (Till 2018). The concept was used during the pre-Cold War period when naval forces grew as a compelling state tool to coerce actors (Le Mière 2011b). Naval diplomacy does not occur during wartime; it is a strategy used during peacetime to achieve certain political ventures through a state's hard power assets (Davidson 2008). The goals of employing naval diplomacy may vary based on the case but would be expected to achieve certain political purposes, such as power projection, compelling rivals, or affecting policies (Rowlands 2018; McConnell and Kelly 1973). As scholars in the past tended to include a divergent list of naval operations as naval diplomacy, it has been challenging to pinpoint an exact definition that is conclusive of the true nature of gunboat diplomacy. Cable's definition, for example, has attracted strong criticisms recently due to his gunboat diplomacy definition incorporating naval forces for practically any policy goal as long as it utilizes the hard power asset of navies (Cable 1994). Earlier studies on this topic can be traced to the works of Alfred Thayer Mahan. Arguing in the context of sea power between 1660–1783, Mahan concluded that a state advances its sea power to protect one's vital national interests, such as the protection of its commerce (Mahan 1898). Mahan further argued that for that reason, states would tend to surge their maritime capabilities to project their naval powers globally. Navies inevitably acted as a consistent policy-coercing tool in the past. Still, in the following years, scholars started to perceive diplomacy at sea through the lens of its diplomatic utility.

At the 20th century, academics started to look at the diplomatic functions of maritime diplomacy as a whole. Luttwak, for example, studied the importance of sea power in general and the diplomatic functions that can assist in excelling a state's national interests (Luttwak 1974). Assumptions regarding the diplomatic utility of maritime diplomacy started when criticisms concerning the growing irrelevance of naval forces in a globalized society began to emerge (Booth 1985; Young 1974; Burhanuddin et al. 2021). Sea power, a term representing a state's assets at sea, could not simply be defined by possessing naval forces alone. Scholars started to view the importance of alternative, non-navy stakeholders that could represent a state's sea power. The term 'maritime diplomacy' then gained traction to explain a state's diplomatic strategies using its maritime-based sources.

Le Miere's *Maritime Diplomacy in the 21st Century* set the tone for deeper academic analysis of the holistic categorization of a state's sea-based power used for diplomatic purposes. The terms gunboat and naval diplomacy started to lose their edge due to evolving international laws and the perception of those hard power assets being too coercive if operated on

large scales. Le Miere stated that maritime diplomacy is the management of international relations at sea and that different actors can carry out this job (Le Mière 2014). He argued that using maritime constabulary forces including coastguards, maritime law enforcing agents, and fishing militia can be defined as 'paragunboat diplomacy' (Le Mière 2011a). The term paramilitary indicates the crew's military-trained background and gunboat to represent how states can utilize these maritime forces in a limited coercive way, such as to compel adversaries.

Le Miere's work in 2014, though impactful in the study of maritime diplomacy, was not the first to capture the importance of non-navies. Coastguards have been an extremely explored actor by maritime diplomacy scholars throughout the past two decades. Several studies highlighted the coastguard's importance in producing cooperative norms among states (Paleri 2015; Greenwood and Miletello 2022). Kelly specifically argued the importance of cooperative norm construction in the Indo-Pacific region, which has gained prominence in the past decade (Kelly 2014). In general, discourses in the coastguard relates to their mandate of safeguarding the state against maritime threats. This is why past scholars argued that coastguards hold a vital mandate to counter non-traditional security threats, such as transnational crimes (Llewelyn 2016; He 2009). However, scholars are starting to perceive that coastguards may have other distinct diplomatic roles besides the pragmatic functions of countering crimes and law violators at sea. Aside from Le Miere, Kim also discussed how coastguards perform the function of countering traditional maritime concerns, such as protection over sovereign claims at sea as the front line of a state's defense (Kim 2018).

This article helps to fill the novelty gaps in the literature on maritime diplomacy. The contributions are twofold; firstly, it discusses discourse in paragunboat diplomacy. Existing scholarship has not engaged in the critical question under the maritime diplomacy discourse of why states adopt paragunboat diplomacy, especially in disputed waters. In the past, besides Le Miere who coined the term in 2014, only Basawantara used paragunboat diplomacy to reflect the growing non-military utilization of maritime forces (Basawantara 2020). Before that, Le Miere used the term 'maritime paramilitaries' to showcase the rise of its use as a foreign policy tool in east Asia (Le Mière 2011a). Although there is a rising acknowledgment that maritime constabulary forces such as coastguards hold impactful diplomatic functions, other actors, such as fishing surveillance vessels, are ignored in the literature on maritime diplomacy.

The second contribution of this article is an empirical investigation of Vietnam's constabulary forces. Existing literature on Vietnam's maritime diplomacy highlights Vietnam's coercive history in disputed waters, such as the South China Sea (de Gurung 2018; Sangtam 2021; Pinotti 2015). Nevertheless, no existing study examines Vietnam's adopted alternative maritime diplomatic strategy in the face of its disputed waters with Indonesia. In addition to the contribution of understanding paragunboat diplomacy, this research also clarifies the role maritime constabulary forces engage in as a diplomatic tool for states. A focus on Vietnam's maritime constabulary forces thus goes against certain assumptions, such as those of Rao, who argues that policymakers can only enhance maritime security through actors such as navies, marine police, and coastguards (Rao 2010). This article asserts that with the growing sensitivity of maritime-related issues, states are cautious of the maritime assets they use and thus prioritize non-military fleets to showcase a mix of cooperative and coercive positions in disputed waters.

## 3. The Growing Strategic Importance of Southeast Asian Coastguards

The rising importance of maritime constabulary forces is most evident in Southeast Asia. For the past two decades, Southeast Asian states aimed to excel in the capacity of respective constabulary forces amid rising maritime-related concerns (Till 2022; Parameswaran 2019). The investments and developments of maritime constabulary forces have focused on empowering coastguards, maritime fisheries vessels, and other law enforcement agencies.

In comprehending those developments, it is vital to consider the importance of the seas for Southeast Asian states. Located at the heart of the Indo-Pacific region, countries in Southeast Asia are mindful of the challenges faced by both traditional and non-traditional maritime security threats (Darwis and Putra 2022; Llewelyn 2016). Southeast Asia is exposed to the contemporary maritime border tensions of the South China Sea and is home to one of the most strategic chokepoints in global trade—the Malacca Straits. The rising importance of the Asian market, primarily due to the rising prominence of China globally, makes Southeast Asian waterways strategically vulnerable to power projections. It is also predicted that commerce through the Malacca Strait will double in the upcoming decade (Kaur 2018).

Indo-Pacific policymakers, especially in Southeast Asia, are aware of these ongoing disputes and dynamics in the Indo-Pacific. Arguably, the importance of the sea has been a common feature in most Southeast Asian states throughout history. Singapore, for example, established its coastguard in 1866 as one of the earlier Southeast Asian nations to perceive the importance of coastguards (Morris 2017). The Philippines coastguard was established a century ago, indicating the strategic importance of enforcing the law at sea (de Castro 2022).

The intensification of Southeast Asian maritime constabulary forces can historically be tracked with China's power projections in the South China Sea since the early 21st century. China's claims in the South China Seas overlap the jurisdiction of half of the ASEAN member states. Evolving discourses in the Belt Road Initiative and China's perception of the importance of securing global shipping lanes in response to the surge of commerce in Asia have repositioned the maritime agenda as a top priority for Southeast Asian states (Hu 2019; CIMB 2018; Zhou and Esteban 2018). Le Miere argued that utilizing constabulary forces is a strategic way of enforcing the law at sea and safeguarding one's sovereign claims (Le Mière 2014).

Constabulary forces, whether coastguards, maritime law enforcement agencies, or maritime militia, have been the Southeast Asian answer to the growing presence of belligerence at sea. To safeguard their claims, Vietnam, the Philippines, Malaysia, and Indonesia have shown clear signals of increasing presence at sea through their constabulary forces. The strategies they incorporate mimic China by using coastguards, maritime militia, and other non-military vessels around disputed seas to solidify claims over disputed waters. Parameswaran contends that the Southeast Asian strategy has been to either construct or solidify their maritime law enforcement agencies as a non-military alternative to the escalation-prone use of state navies (Parameswaran 2018). Vietnam and Indonesia, for example, established their coastguard agencies in 2013 and 2015, respectively, while Malaysia and the Philippines ramped up their investments in coastguards (Parameswaran 2019).

A common feature of Southeast Asian maritime constabulary forces is their divergent mandates specific to the navies. Traditionally, Southeast Asian coastguards gained considerable attention from policymakers after the September 11 attacks. Malaysia, for example, established its coastguard in 2005 to respond to the growing transnational threat of terrorism (Tarriela 2022). Maritime law enforcement agencies have also been utilized for countering non-traditional security threats, such as transnational crimes, smuggling, and IUUF. Contemporary developed constabulary maritime forces in Southeast Asia still accommodate the discourse of enforcing the law at sea to counter those non-traditional security threats, but also solidify a state's territorial claims at sea through the presence of non-military vessels.

An indication of the rising importance of these maritime constabulary forces is also represented through the acquisitions made. Vietnam's coastguard, previously known as the Vietnam marine police until 2013, currently holds the most significant maritime vessel fleet for patrolling functions of the coastguard. Its DN-4000, a 4300-ton vessel, is Southeast Asia's largest coastguard vessel, reflecting Vietnam's perception of the rising importance of securing the sea (Vu and Phuong 2017). Larger coastguard fleets are a shared priority for other well-established maritime law enforcement agencies in Southeast Asia. Indonesia is

another example, employing a 2400-ton KN Tanjung Datu vessel commissioned in 2017 (BAKAMLA 2020).

Investment in larger maritime vessels indicates Southeast Asian states' strategic intentions to project power through their maritime constabulary forces. Maritime constabulary forces should be employed by Southeast Asian states to enforce the law internally within the state borders. However, taking the example of Indonesia's BAKAMLA, these agencies are now mandated to secure the sovereignty of states at sea (BAKAMLA 2021). Navies are no longer seen as maritime diplomatic assets that can pragmatically tackle the issues of non-traditional maritime threats that do not require such a heavily-armed asset to be deployed. As stated previously, this reflects the notion of mandating an extensive list of tasks to these newly emerging agencies to have the strategic edge of feasibly being employed in practically any possible scenario. The flexibility maritime constabulary agencies generate is why Southeast Asian states have shown this trend of increasing their capacities at sea via non-military means.

In connection to the literature review on paragunboat diplomacy, coastguards and other maritime law enforcement agencies will not cause the same level of escalation as navies. Parameswaran suggested the possibility that the number of incidents related to the use of coastguards has increased, exposing the possibility of a 'lower threshold of escalation' (Parameswaran 2019, p. 7). However, in reality, the tensions we have seen between maritime constabulary agencies and maritime militias are not at the level that could cause a diplomatic crisis among states.

Being exposed to China's nine-dash line, Vietnam's development of maritime constabulary forces has been noticeable in the past decade. Not only has Vietnam attempted to project its power at sea through larger patrolling vessels for its coastguards, but Vietnam also continues to use its VFRS to consolidate its claims in the Natuna Seas. As shown in the cases of VFRS-accompanied IUUF and suspicious maneuvers in the seas between 2021–2022, Vietnam's multi-directional position on the South China Sea considers the importance of maritime constabulary forces as an integral part of its South China Sea strategy.

## 4. Vietnam's Power Projection and Coercive Posture in the Natuna Seas

Vietnam perceives that protecting its sovereignty in the South China Sea is obligatory for its domestic population. In understanding the national security strategy of a country such as Vietnam, the strategies employed can be traced from the political platforms of the Vietnamese Communist Party. The core of its national security is based on the concept of "Fatherland Protection" (*bao ve Tu quoc*), which focuses on the importance of safeguarding Vietnam's independence, sovereignty, and socialist regime (Luving 2018). It did not come as a surprise, therefore, that in Vienam's latest defense white paper published on 25 November 2019 that the Vietnam Ministry of National Defense referenced the South China Sea as its major security threat (MOND 2019).

China's growing assertiveness in the South China Sea has made it difficult for Vietnam to cope. As of 2022, China remains Vietnam's second leading two-way trade partner (OEC 2022). As seen in Figure 1, China exhibits greater economic importance for Vietnam, as the two-way trade was valued at USD 7.39 billion in 2018. Nevertheless, the overlapping EEZ claims between China and Vietnam make cutting-off diplomatic ties or adopting a coercive posture difficult. Existing literature argues that Vietnam's alignment position is concerned with China's rise in the region and its growing belligerence on maritime border tensions (Medeiros 2010; Goh 2016, 2007; Kuik 2021).

Overlapping EEZ claims also apply to Indonesia and Vietnam. Indonesia's portion of the North Natuna Seas coincide with the EEZ of Vietnam, which has aggravated bilateral relations between the two Southeast Asian powerhouses. Efforts towards reconciliation were introduced in 2003 when both agreed to sign an agreement outlining their continental shelf boundaries. Nevertheless, the inconclusive EEZ boundaries have led to several limited tensions. Between 2017–2019, Vietnamese and Indonesian coastguards interchangeably exchanged assertive maneuvers to enforce sovereignty in the Natuna Seas (Phan 2021).

It is difficult to pinpoint a precise definition and parameter of assertive maneuvers, as it tends to depend on how policymakers define a given crisis. Nevertheless, this article will focus on cases framed by Indonesian policymakers as an assertive or provocative maneuver conducted by Vietnamese officials at sea, usually called *provokatif* or berbahaya. Jokowi's former minister of marine affairs and fisheries, Susi Pudjiastuti, claimed to have bombarded 558 fishing vessels that illegally intruded Indonesian waters, most of which were Vietnamese fishing vessels (Purba 2019). Indonesia and Vietnam's continuously conflicting legal stances have allowed fishermen to operate beyond EEZ borders, which continues to cause tensions between Jakarta and Hanoi.

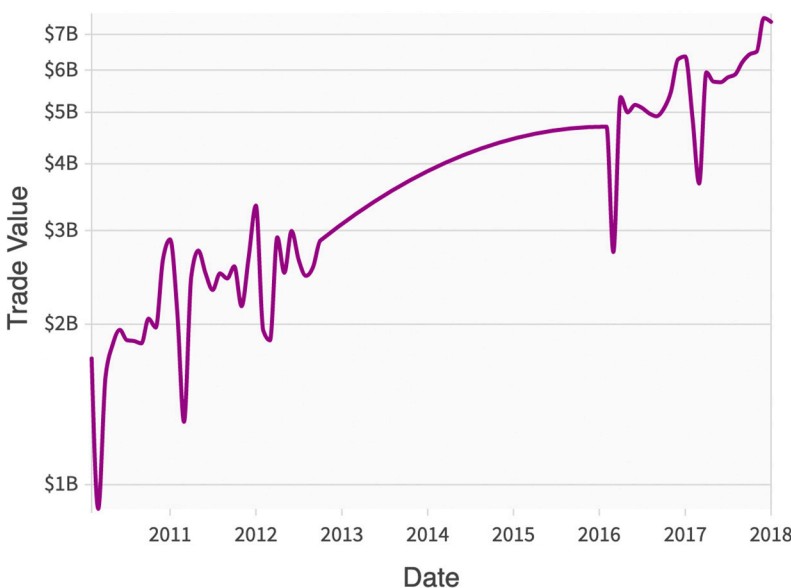

**Figure 1.** Trade between Vietnam and China (2010–2018). Source: Observatory of Economic Complexity (OEC 2022).

The bilateral suspense faded after Indonesia and Vietnam agreed on an EEZ demarcation agreement on 22 December 2022. The agreement took 12 years of negotiation to conclude, indicating the substance of the maritime borders for both Jakarta and Hanoi (Stranjo 2022). Considering both parties' rising IUUF, the demarcation agreement is hoped to provide clarity and collaboration for any actions taken in the overlapping EEZ areas. Recent developments of arrests and clashes between constabulary forces from both countries are expected to end after the conclusion of this agreement. The agreement projects Vietnam's intentions to adopt cooperative measures to solve the sensitive issue of maritime boundaries.

However, due to the sensitivity of Southeast Asian states to maritime-related boundaries, Indonesia and Vietnam continue to project sea power through the utilization of both military and non-military posturing at sea. Despite a cooperative stance in the EEZ demarcation agreement, Vietnam does not indicate that power projection should also be eliminated in the Natuna Seas. In reality, just months before the conclusion of the demarcation agreement, both sides constantly clashed in the EEZ boundaries of the North Natuna Seas (IOJI 2022b). Vietnam perceives that the risk of being absent in tensioned waters leads to losing sovereignty. Similar to how China viewed the Declaration on the Conduct of Parties in the South China Seas (DOC), Vietnam will not abandon its power projection in the North Natuna Seas.

In comprehending Vietnam's power projection in the Natuna Seas, one must revisit Vietnam's contemporary stance concerning the South China Seas. After the standoff between China and Vietnam following the 2014 Chinese drilling rig Hai Yang Shi You 981 crisis, Vietnam regarded maritime border tensions as a top foreign policy priority (Storey 2014). Since then, Vietnam has known that states will not abide by UNCLOS

regarding maritime boundaries and border contestations. Though the 2014 tensions quickly faded, China's ambitious reclamation in the Spratly and Paracel Islands, as well as its employment of militarized fishing fleets throughout the overlapping EEZ claims in the South China Sea, all contributed to Vietnam's coercive turn in their South China Sea posturing, including in the Natuna waters. As Gilang contends, after the incidents in 2014, Vietnamese policymakers committed to enforcing the law at sea by updating fishing fleets and enhancing the capacity of its coastguards (Giang 2018).

Vietnam's coercive turn amid China's growing assertiveness in the South China Sea should attest to why Vietnam chooses to project power in the North Natuna Seas through its maritime constabulary forces. In 2009, Vietnam concluded its law on militia and self-defense forces, allowing the operationalization of its fishing militia (Phuong and Vu 2017). In recent years, Vietnam's maritime constabulary forces (coastguards and fishing surveillance fleets under the Vietnam Ministry of Agriculture and Rural Development) received equipment and technical assistance from India, Japan, and the US (Giang 2018). A series of newly introduced laws to enhance Vietnam's maritime constabulary forces and enhance its capacities reflects Vietnam's determination to solidify its claims at sea.

Vietnam also underwent reclamation of the Spratly Islands to showcase its decisive stance in the face of maritime border tensions. As reported by the Asian Maritime Transparency Initiative, Vietnam continues to upgrade its island bases through reclamation and construction. In the past two years, for example, the West Reef and Sin Cowe Island have been the focus of Vietnam's power projection through the placement of communication towers, administration buildings, and coastal defense technologies (AMTI 2021). Vietnam's decisive South China Sea policy in the Spratly Islands seems to likewise take place in the Natuna waters but through the utilization of its maritime constabulary forces.

## 5. Vietnam's Paragunboat Diplomacy in the North Natuna Seas

This article argues that Vietnam's use of constabulary forces (coastguards, fishing surveillance vessels) is a form of Vietnam's paragunboat diplomacy. Paragunboat diplomacy is maritime diplomacy that utilizes non-military equipment at sea to exert its influence. As described in this article, paragunboat diplomacy's rising prominence is basically because of the tactical flexibility of using constabulary forces that only leads to limited tensions at sea. However, Vietnam wishes to solidify sovereign claims due to the active presence of state agencies.

The increasing presence of Vietnam's constabulary forces in the North Natuna Seas coincides with the surge of Vietnamese IUUF. As shown in Figure 2 and based on satellite imagery, most of the Vietnamese IUUF in the North Natuna Seas are conducted within Indonesia's EEZ, with some approaching deep into Indonesia's territorial seas. Data from the Indonesia Ocean Justice Initiative show that the number of illegal Vietnam fishing vessels intruding into Indonesian waters accumulated to more than 200 cases in 2021 (IOJI 2022d).

A surprising revelation regarding Vietnamese IUUF in the Natuna Waters is that, in most cases, they were accompanied by VFRS. Darwis and Putra (2022) argued that in five specific cases between 2020–2021, Vietnamese IUUF boats were safeguarded by VFRS, suggesting that the intrusions were conducted intentionally (Darwis and Putra 2022). In that study, Darwis and Putra found that all apprehensions were challenging to conduct due to the obstructions by VFRS acting to guard those fishing vessels against being captured or processed by the Indonesian Navy, BAKAMLA, or patrol vessels from the Indonesian Ministry of Marine Affairs and Fisheries. The use of VFRS to guard fishing vessels and conduct power projection maneuvers at sea is known as Vietnam's paragunboat diplomacy in the North Natuna Seas.

In 2022 alone, the presence of VFRSs continues to surge, leading to daily abruptions to local Indonesian fishermen. As seen in Figure 3, between July and September 2022, VFRS continues to conduct provocative maneuvers by navigating directly into Indonesia's EEZ to project its power at sea (IOJI 2022b). Before that, between June and July 2022, as

seen in Figure 4, the same maneuvers were conducted by Vietnamese VFRS, showing its intentionality in navigating Indonesian waters (IOJI 2022a). These developments show that Vietnam's employment of VFRS as a maritime constabulary force is Vietnam's most effective tool in asserting its sovereign claims near Indonesia's North Natuna Seas through conducting patrols and showing its presence.

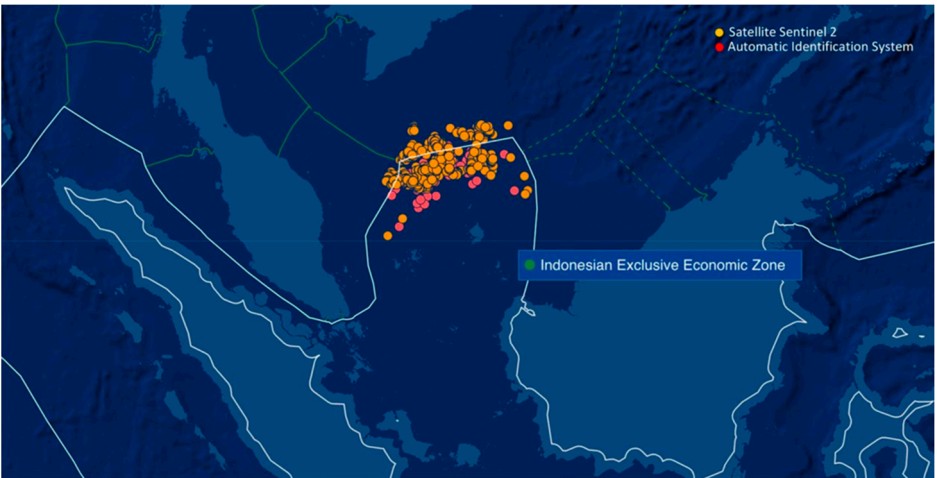

**Figure 2.** Areas of Vietnamese IUUF in the Natuna Waters in 2021 (based on satellite Sentinel-2 and an automatic identification system). Source: Indonesia Ocean Justice Initiative (IOJI 2022d).

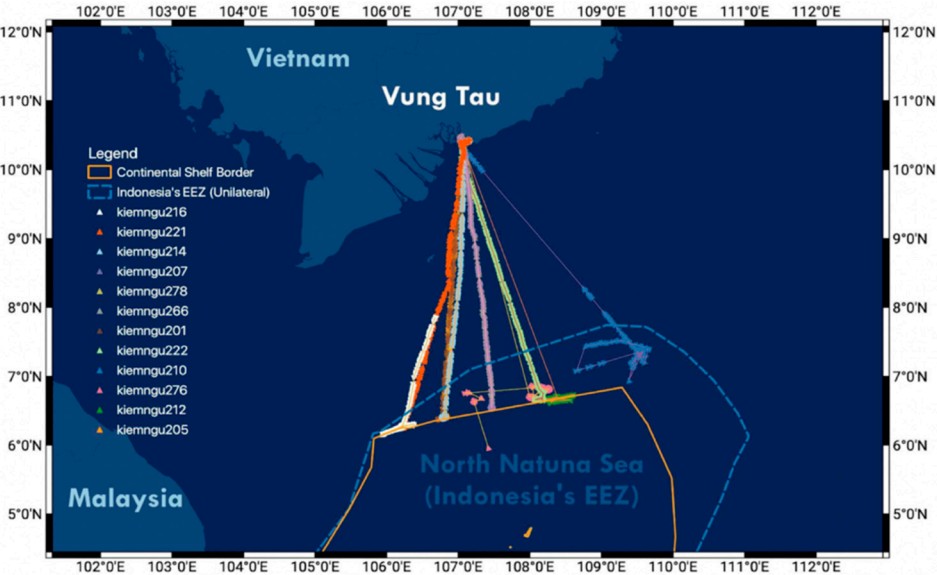

**Figure 3.** Track of VFRS between July and September 2022. Source: Indonesian Ocean Justice Initiative (IOJI 2022b).

Since 2022, Vietnamese paragunboat diplomacy operating within the continental shelf boundary line has steadily increased. Before June 2022, four VFRS vessels were also seen within 7 to 10 nautical miles of Indonesia's continental shelf boundary line, the same maneuvers conducted throughout 2021 (IOJI 2022a, 2022c). VFRS as a maritime constabulary agency is used to escort Vietnamese fishing boats operating IUUF in the North Natuna Seas and, together with the VFRS, it represents a power projection through non-military means. This development is not common, as Vietnam's utilization of the VFRS is supposed to counter cases of illegal fishing. Still, in most cases in 2021–2022 they were ignorant and allowed such conduct to occur in Indonesia's EEZ. The act of escorting Vietnamese fishermen is not a coincidence but a strategically calculated policy of

paragunboat diplomacy due to the tactical flexibility and numerous advantages of using maritime constabulary forces to assert sovereign claims.

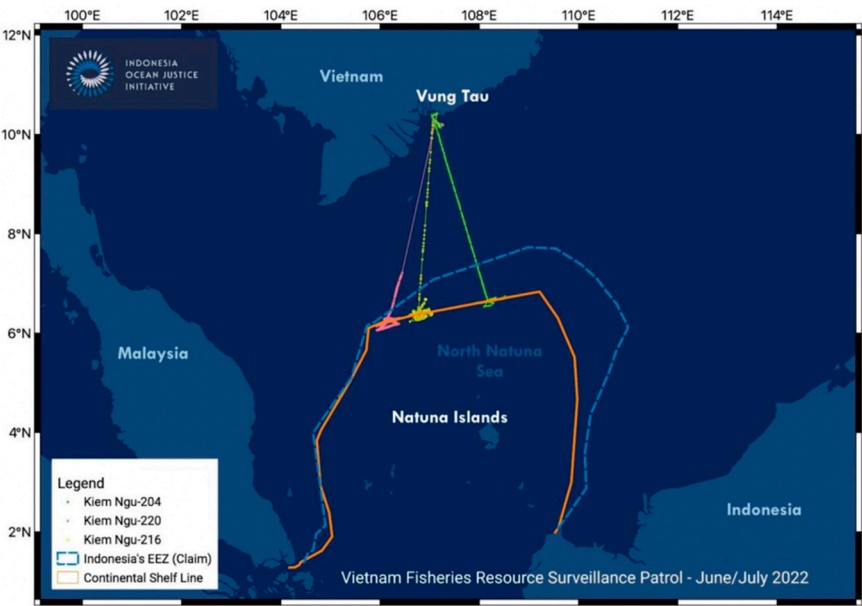

**Figure 4.** Track of VFRS between June and July 2022. Source: Indonesian Ocean Justice Initiative (IOJI 2022a).

Vietnam's maritime constabulary forces violate the 'due regard' clause 234 of UNCLOS. Due regard demands coastal states balance freedom of navigation and preserving the marine environment (Solski 2021). Unfortunately, in the case of VFRS in the North Natuna Seas, it was evident that Vietnam did not respect its due regard responsibilities towards Indonesia's sovereignty. As in the case of the South China Sea tribunals of 2016, the Chinese government was found guilty of allowing Chinese-flagged vessels to fish in the Philippines EEZ and it was viewed as a direct action from the Chinese government (not independent fishermen) (Noyes 2016). In relation to the case of Vietnam's intrusions, it was reported that the VFRS had been allowing illegal fishing of Vietnamese fishermen in the North Natuna Seas, and in some cases, even protected these ships against the pursuits of Indonesian maritime constabulary forces (IOJI 2021, 2022a, 2022b, 2022c, 2022d). It is difficult to argue that Vietnamese officials were unaware of the IUUF taking place in Indonesia's EEZ, thus indicating that Vietnam violated its due regard responsibilities to Indonesia.

Vietnam's coercive turn to paragunboat diplomacy cannot be analyzed independently from Vietnam's South China Sea stance. For Southeast Asian states, resolving maritime boundary tensions in the South China Sea would take decades to resolve. Peaceful maritime diplomatic strategies are not sustainable and will only clear the path for other aggressors to exert their claims in the contested seas. It is not ideal for Vietnam to adopt a coercive stance against China or Indonesia, as they are both strategic partners for Hanoi. If Vietnam were to showcase its claims in the North Natuna Seas through its navy, it would ignite a signal of coerciveness to Indonesia and an escalation of the conflict could quickly unravel.

The use of the Vietnam People's Navy contains unprecedented risks for Hanoi. A sudden development or decisive gesture in the North Natuna Seas by the Vietnamese Navy would lead to militarizing maritime border tensions. Vietnam is wary that Jakarta is sensitive towards any maritime contestations it confronts. For example, as a response to the growing assertiveness of China in the Natuna Seas, Jokowi responded by conducting a cabinet meeting above Indonesia's warship in 2016 and a year later remained in the Natuna Waters to the North Natuna Seas (Kapoor and Jensen 2016; Connelly 2017). A series of maritime-related confrontations and challenges have led Jakarta to adopt a cautious posture, ready to strike back at any assertive maneuvers made. Even with the conduct of

VFRS to accompany Vietnamese fishermen, Indonesia's BAKAMLA has not hesitated to respond decisively (BAKAMLA 2021; Fadli 2020a, 2020b).

Maritime diplomacy commonly involves coercive, persuasive, and cooperative means (Le Mière 2014). Persuasive maritime diplomacy through the use of the Vietnamese navy may ignite unprecedented consequences due to the peculiar nature of the maneuver. As Vietnam cannot afford to undertake a coercive maritime strategy due to its interest in safeguarding relations with Indonesia, Vietnam's remaining option is through cooperative means. As seen in the December 2022 agreement between Jakarta and Hanoi, it is still possible to agree upon an EEZ demarcation agreement. However, this article argues that its impact would be similar to the DOC. Southeast Asian states consider maritime borders contentious, especially for their domestic constituents. An agreement concluded 12 years after the rise and fall of Vietnam and Indonesia's maritime bilateral relations would hardly make a significant difference to the maritime diplomatic strategy of Vietnam.

Thus, Vietnam's paragunboat diplomacy through maritime constabulary forces aims to consolidate sovereign claims without escalating conflict. Le Miere introduced this idea, arguing that the strength of paragunboat diplomacy lies in the exhibition of de facto sovereignty (Le Mière 2014). As with Vietnam's reclamation and construction of smaller islands in the Spratly Islands, Indonesia's Natuna strategy is to demonstrate Vietnam's active presence in the contested waters. Through effective occupation, Hanoi aims to normalize its claims and further push back the boundary lines between Indonesia and Vietnam.

Furthermore, Vietnam's paragunboat diplomacy generates maximal results, offering military flexibility without resorting to conflict escalations. Vietnam is highly cautious of the moves it displays in contested waters. Vietnam is not a middle power, as is the case with Indonesia, nor an emerging power such as China. It faces difficulties establishing domestic order and defining its position in world affairs (Do 2022). Paragunboat diplomacy offers Vietnam the flexibility to raise tensions and de-escalate in minutes. Vietnam can adopt pressure and back off when it is no longer effective. The military equipment within VFRS maritime constabulary vessels can undergo operations in vast geographic areas and choose between offensive–defensive posturing (or both). Therefore, Hanoi does not need to commit to a long-term strategy as long as its maritime constabulary forces can remain present in the disputed waters. The fact that these constabulary forces are not under the command of the navies indicates that they are, in reality, civilian vessels. In the North Natuna Seas, the only level of escalation between Jakarta and Hanoi that can be reached is capturing boats and crew members, ramming ships, and using a highly limited armory. The chances of causing deaths and diplomatic crises are slim to none.

## 6. Conclusions

The North Natuna Seas remain a turbulent disputed territory for Indonesia and Vietnam. Growing maritime tensions between Jakarta and Hanoi have surged with the rise of Vietnamese-flagged IUUF into the Indonesian EEZ and the presence of Vietnam's maritime constabulary forces, VFRS. This article argues that Vietnam's response to the inconclusive maritime dispute between Indonesia and Vietnam in the Natuna waters included the utilization of paragunboat diplomacy to showcase Vietnam's power in the North Natuna Seas.

The use of maritime constabulary forces and coastguards has risen in recent years. Southeast Asian states, especially disputants to the South China Sea, show similar patterns of concern regarding China's nine-dash line. The actual dilemma that policymakers in Southeast Asia face is that they cannot afford to respond to China with hostility due to the importance of Beijing in their economies. Vietnam is no different, as it adopts a mixture of cooperative and coercive maritime positions in a desperate attempt to project power.

In the case of the Natuna Seas, Vietnam continues to adopt its traditional, limited-coercive maritime diplomatic strategy through paragunboat diplomacy. The development of Vietnam's coastguard and other maritime law enforcement agencies represents Vietnam's intent to solidify its sea-based sovereignty through non-military measures. The article shows that Vietnam employs the VFRS to occupy the disputed waters effectively. It enjoys the tactical military flexibility that maritime constabulary forces generate, and the non-escalatory means to solidify sovereign claims make it a consistently adopted maritime diplomatic strategy for Hanoi. Between 2021 and 2022, the suspicious maneuvers of VFRS and the continuous safeguarding of Vietnamese IUUF fishing boats in Indonesia's EEZ are desperate attempts to excel Vietnam's maritime claims.

**Funding:** This research received funding from the Indonesian Endowment Scholarship Fund of Indonesia, grant number PER-4/LPDP/2021.

**Informed Consent Statement:** Not applicable.

**Acknowledgments:** The author would like to acknowledge the support given by the Indonesian Endowment Schol-arship of Indonesia (LPDP) and Universitas Hasanuddin, for supporting this research.

**Conflicts of Interest:** The author declares no conflict of interest.

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
