# Peer review of "Rise of Constabulary Maritime Agencies in Southeast Asia: Vietnam’s Paragunboat Diplomacy in the North Natuna Seas"

_socsci, doi:10.3390/socsci12040241_

Round 1

Reviewer 1 Report

The paper provides an interesting subject matter and the theme is important regarding the regional security of Southeast Asia. The paper may provide a good contribution to the field since it has the potential to provide a more detailed analysis of Vietnam's paragunboat diplomacy and how this affects regional security dynamics. 

However, as it is in its current form, the paper has not yet reached a level that allows for a significant contribution. The reviewer highlights a few key issues that merit consideration:

1) Articulate a specific research question. While the paper provides an argument, it should detail the overall problem in which the research question is framed;

2) Justify the methodological choices. In particular, explain the sources and time frame chosen. 

3) Develop the analysis by providing more detail regarding Vietnam's actions. More precisely, frame Vietnam's actions in the context of its overall security strategy - this is necessary to substantiate certain claims made in the paper regarding intentionality. Also, detail and explain the "provocative maneuvers". The author(s) allude to the increase in the number of movements of Vietnamese vessels, but the analytical detail is absent, as is a qualification of these activities.

4) Review the English. The text is ridden with grammatical errors.

Reviewer 2 Report

The thesis of this article is intriguing and asserts that the paragunboat diplomacy is the best option for Vietnam in regards to the North Natuna Sea. The historical usage of "gunboat diplomacy" occurred in the early 20th century imperial/colonial  ambitions of the US.  It was the intimidation factor of the gunboat diplomacy that the US used in order not to cause bloodshed but asserts its power. Nonetheless, it angered other nations while accomplishing its intended purpose. Thus, the usage of a paragunboat diplomacy seems to weave in a complex history of diplomacy by intimidation, colonialism, and political relations. It implies that civilians have the support and respect of its government to act on its behalf which makes it quite intriguing. The assertion that there will be no death or diplomatic crisis from these tactics may be too naive. It seems that the ramming of boats and vigilantes at sea may cause more violence that could lead to a diplomatic crisis, especially when trying to keep military and political powers out of direct contact. This raises an interesting dilemma that will need further research and investigation. 

Are civilians given such power to speak authoritatively for the government? Do government only use its citizens as pawns rather than its direct charge, especially in countries that are more authoritative and centralized? 

These are just questions the reader thought about while reading through the paper. The paper has a strong argument and has defended it well. It will add to the broader discussion of "paragunboat diplomacy" in light of the ongoing conflict in the South China sea region and beyond. 

Round 2

Reviewer 1 Report

Having read the article, the author(s) addresses the main concerns raised in the initial review and I believe it is improved and merits publication.